# CRISPR gRNA phenotypic screening in zebrafish reveals pro-regenerative genes in spinal cord injury

Marcus Keatinge[1]◔*, Themistoklis M. Tsarouchas[1]◔, Tahimina Munir[1], Nicola J. Porter[1], Juan Larraz[1], Davide Gianni[2], Hui-Hsin Tsai[2], Catherina G. Becker[1‡], David A. Lyons[1‡], Thomas Becker[1‡*]

**1** Centre for Discovery Brain Sciences, University of Edinburgh, Edinburgh, United Kingdom, **2** Biogen, Cambridge, Massachusetts, United States of America

◔ These authors contributed equally to this work.
‡ These authors are joint senior authors on this work.
* mkeatin2@exseed.ed.ac.uk (MK); thomas.becker@ed.ac.uk (TB)

**Data Availability Statement:** All relevant data are within the manuscript and its Supporting Information files.

## Abstract

Zebrafish exhibit robust regeneration following spinal cord injury, promoted by macrophages that control post-injury inflammation. However, the mechanistic basis of how macrophages regulate regeneration is poorly understood. To address this gap in understanding, we conducted a rapid *in vivo* phenotypic screen for macrophage-related genes that promote regeneration after spinal injury. We used acute injection of synthetic RNA Oligo CRISPR guide RNAs (sCrRNAs) that were pre-screened for high activity *in vivo*. Pre-screening of over 350 sCrRNAs allowed us to rapidly identify highly active sCrRNAs (up to half, abbreviated as haCRs) and to effectively target 30 potentially macrophage-related genes. Disruption of 10 of these genes impaired axonal regeneration following spinal cord injury. We selected 5 genes for further analysis and generated stable mutants using haCRs. Four of these mutants (*tgfb1a*, *tgfb3*, *tnfa*, *sparc*) retained the acute haCR phenotype, validating the approach. Mechanistically, *tgfb1a* haCR-injected and stable mutant zebrafish fail to resolve post-injury inflammation, indicated by prolonged presence of neutrophils and increased levels of *il1b* expression. Inhibition of Il-1β rescues the impaired axon regeneration in the *tgfb1a* mutant. Hence, our rapid and scalable screening approach has identified functional regulators of spinal cord regeneration, but can be applied to any biological function of interest.

## Author summary

Nerve connections that are severed in spinal cord injury do not heal, which can lead to permanent paralysis. Lack of repair may in part be due to prolonged inflammation of the injury site. In contrast, zebrafish show excellent repair of nerve connections after spinal injury and this is associated with controlling inflammation. Due to recent advances in genetic technology (CRISPR/Cas9) we can now determine the function of genes that

**Funding:** This work was supported by a Wellcome Trust (https://wellcome.org/) Senior Research Fellowship (102836/Z/13/Z) to DAL and by Biogen (https://www.biogen.com/en_us/home.html) who provided funding via a scientific research agreement with DAL. Work in the Becker group is funded by the Era-Net Neuron Cofund consortium NEURONICHE (https://www.neuron-eranet.eu/) administered by the MRC (https://mrc.ukri.org/) under grant number MR/R001049/1 with contributions from MRC, Spinal Research (https://spinal-research.org/) and Wings for Life (https://www.wingsforlife.com) to CGB, as well as project grants from the BBSRC (https://bbsrc.ukri.org/) to TB (BB/R003742/1) and EPSRC (https://epsrc.ukri.org/) to CGB (EP/S010289/1). The funders had no role in study design, data collection and analysis, decision to publish, or preparation of the manuscript.

influence regeneration in the living zebrafish in a matter of days. Here we devise a very rapid screening method for the function of inflammation-related genes in zebrafish larvae after spinal cord injury. We find a number of genes that are necessary for repair of nerve connections and control of the inflammation after injury. This provides important leads to improve our understanding of the role of inflammation in spinal cord injury. Moreover, our fast and robust screening method can be adopted by other researchers to screen for gene functions in a whole animal, which was previously not easily possible.

## Introduction

Zebrafish, in contrast to mammals, functionally regenerate axonal connections across the injury site after spinal cord injury. Prolonged inflammation is detrimental to recovery from a spinal injury in mammals, but in zebrafish, pro-inflammatory cytokines are rapidly down-regulated and the immune response generally promotes regeneration [1,2]. Specifically, previous work has shown that the presence of blood-derived macrophages is crucial for axonal reconnection in the injured larval spinal cord and recovery from paralysis [3]. These macrophages control the injury site environment by reducing the number of anti-regenerative neutrophils and by mitigating the injury-induced expression of pro-inflammatory cytokines, such as *il1b* by neutrophils and other cell types. However, the mechanisms and signals by which macrophages keep neutrophils and *il1b* expression in check during regeneration are largely unknown.

To address these mechanisms, we decided to establish a screening pipeline to test how immune system-related genes might affect spinal cord regeneration *in vivo* using larval zebrafish [4–7]. CRISPR-based approaches now allow scalable assessment of gene function in zebrafish, because phenotypes of interest can be observed already in acutely injected, mosaically mutated embryos [8–15]. This has been exploited for phenotypic screening [9,16–18] and improvements in CRISPR/Cas9-based technologies. For example, the use of synthetic RNA Oligo CRISPR guide RNAs (sCrRNAs), has led to the availability of highly efficient gene targeting. However, due to the limited number of sCrRNAs characterised to date, it is unknown whether the rate of highly active sCrRNAs is sufficiently high to use these in phenotypic screening [19,20]. Compensating for that by injecting multiple guides targeting the same gene carries the risk of increasing off-target effects (false-positives) [21–23]. Therefore, we reasoned that developing a pipeline that includes pre-screening of guide RNAs *in vivo* for high activity would allow us to identify and prioritise highly active guides for phenotypic screens. For robust and quick pre-screening, we used a restriction fragment length polymorphism (RFLP) —based approach in which the sCrRNA cas9 cut site overlaps with restriction enzyme recognition sites. This ensures that we retain a high degree of freedom in target selection and scalability of the approach.

Here we demonstrate, by testing 350 sCrRNAs for their efficiency that, although variable, almost half can be classified as being highly active sCrRNAs (haCRs). Pre-screening for haCrs enabled us to effectively target 30 genes of interest for their effect on spinal cord regeneration, and we verified 4 genes as positive regulators of successful regeneration in larval zebrafish. Deleting *tgfb1a* led to prolonged presence of neutrophils and increased *il1b* expression, similar to effects of genetic removal of macrophages. Hence, we identify *tgfb1a* as a signalling molecule that controls inflammation after spinal injury in zebrafish, providing mechanistic understanding of the pro-regenerative role of the immune system in zebrafish. Moreover, we present a

rapid sCrRNA design and pre-selection process that will generally facilitate rapid and robust phenotypic screening *in vivo*.

## Results

To establish a phenotypic screening platform to assess the function of candidate genes in regulating the response to spinal cord injury, we set out to determine the general efficiency of sCrRNAs and number of haCRs that need to be co-injected to obtain a sizeable loss-of-function.

### sCrRNA activity is variable in vivo

We targeted 350 genomic sites (S1 Table) of genes in the general context of nervous tissue injury or disease and determined their activity towards the targeted sites by injecting sCrRNAs into the zebrafish zygote. To do this efficiently, we targeted recognition sites for restriction enzymes, such that we could use resistance to enzyme digestion in restriction fragment length polymorphism analysis (RFLP) of individual embryos as a proxy for mutagenesis activity at the target site (Fig 1A). Activity was determined by measuring band intensities of undigested and digested bands and calculating their ratio. This ratio, from 0 (no activity) to 1 (complete target mutagenisation) was expressed as percentage efficiency. For example, a ratio of 0.9 was thus expressed as 90% efficiency. We defined as a haCR those with an efficiency of > 90%. We favoured recognition sites for *bsl1*, *xcm1* and *bstxi* restriction enzymes, because these enzymes contain the necessary PAM sites for CrRNAs in their recognition sequences. Moreover, the recognition sites of these enzymes are also relatively large (>10 nucleotides), consisting of mostly redundant sequences. This makes them sensitive to formation of insertion and deletion mutations (indels) but also resistant to potential single nucleotide polymorphisms. Although we favoured these enzyme sites, any can in theory be used as long as their motif crosses the Cas9 cut site. For example, we also commonly used *mwoI*, *bsrI* and *bstnI*. We used unmodified sCrRNA, because reportedly there are no systematic differences for chemically modified versions [19].

To increase the likelihood that sCrRNAs were function-disrupting, we preferentially targeted the 5' end of the genes' coding regions, increasing the chance that frameshift-inducing indels would lead to early stop codons likely to severely disrupt protein translation. In cases in which suitable 5' target sites were not available, we targeted established functional domains. The subsequent RFLP analysis indicated that 44% of all tested sCrRNAs were haCRs (Fig 1A). Nonetheless, the activity of sCrRNAs is variable, with many of them showing low activity (e.g. 106 of 350 sCrRNAs showed < 50% activity).

We next wanted to determine to what extent guide RNA mutation rate could be successfully predicted by application of established in silico-based rules to increase the haCr detection rate, which would ideally avoid the *in vivo* testing phase. Using CHOP-CHOP [24], we tested prediction guidelines which assume that GC content, single and dinucleotide identity at each position improves efficiency [25], that cytosine at the -3 position, adenines at -5/-12 and guanines at -14/-17 are favourable for high activity [26], and that guanine enrichment and adenine depletion increases targeting efficiency [27]. Correlation between predictions and actual efficiencies observed in our RFLP-based assays were weak for all of these ($R^2$-values between 0.028–0.044; Fig 1B–1D). Our preference for the 5' end of the gene did also not bias sCrRNAs for high activity (Fig 1E). Hence, we found that conventional design rules do not strongly predict *in vivo* mutation rates for sCrRNAs. This indicates the importance of pre-screening guide RNAs *in vivo* to ensure an adequate mutation rate within injected animals, which the RFLP-based approach allows to be executed rapidly and efficiently.

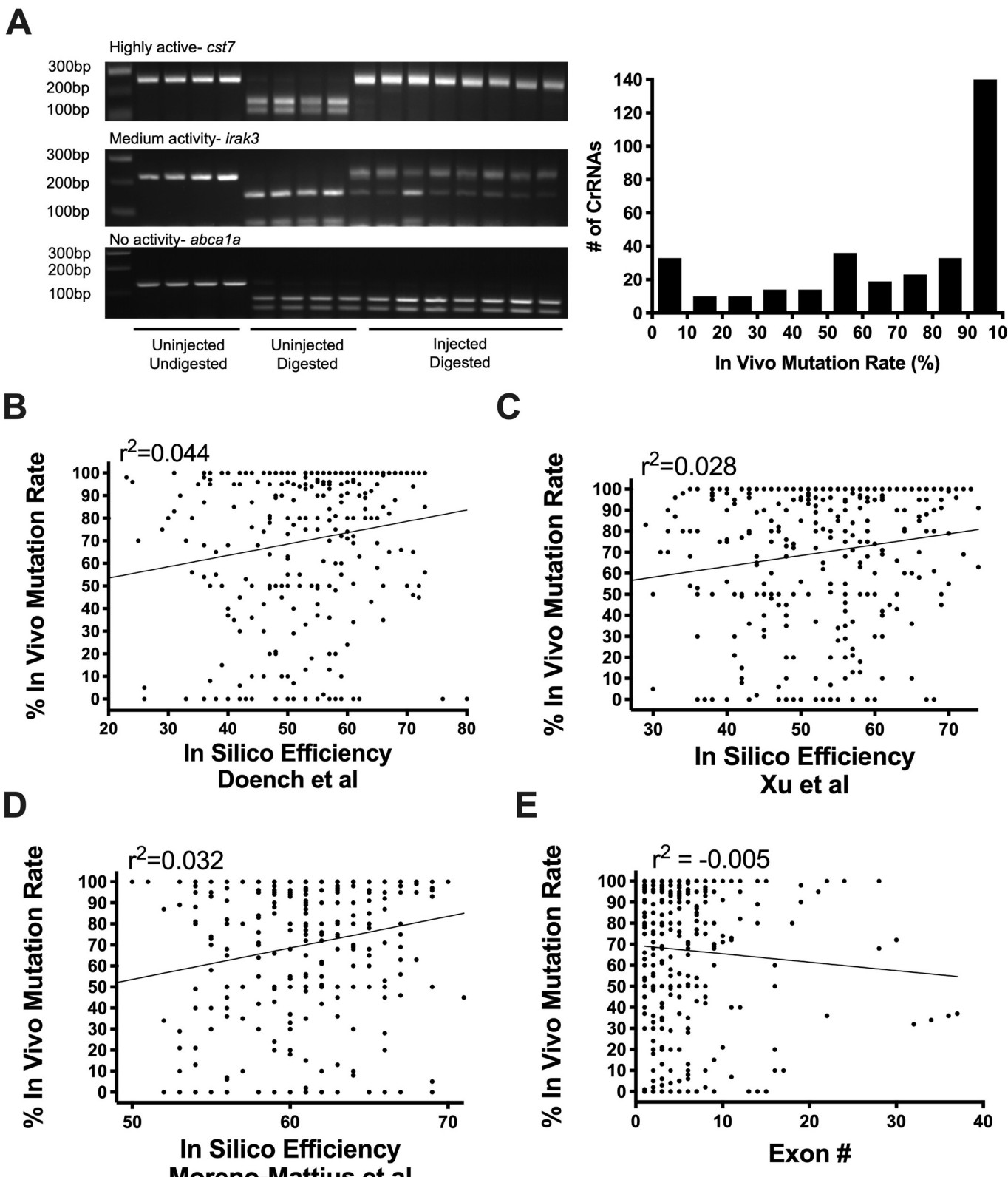

**Fig 1. *In vivo* pre-screening identifies highly active sCrRNAs. A:** Example gels to assess sCrRNA *in vivo* mutation rate by resistance to restriction enzyme digest (RFLP) are shown. These indicate > 90% mutation rate (top), medium mutation rate (middle) and no detectable mutation rate (bottom). Each lane is derived from one animal. The chart shows the distribution of > 350 sCrRNAs by *in vivo* mutation rate. **B-D:** Activities of individual sCrRNAs show weak correlation with predicted in silico efficiency using different prediction rules (see results). **E:** sCrRNA activities are not correlated with their relative position to the start codon *in vivo*.

## haCRs efficiently ablate gene function

For phenotypic screening it is important that levels of functional protein are strongly reduced. This may best be achieved by frameshift mutations that can statistically be assumed for almost 66% of all alleles (1–1/3) for injection of one haCR [27]. We aimed to use 2 haCRs per gene, where available, because the total proportion of frameshifts can be estimated to be 89% [1- (1/3 * 1/3)] per allele, so 79% for bi-allelic frameshifts in a cell (0.89 * 0.89). In addition, also in-frame indels may not be tolerated, particularly when known functional domains are targeted. Designing and injecting more than two haCRs could in principle increase the bi-allelic frameshift rate further, but with ever diminishing returns (e.g. 92% of cells for three haCRs compared with 79% for two), and comes at the cost of increasing the possibility of off-target actions.

We used direct sequencing to test the prediction of substantial frameshift activity by using two haCRs (S1A–S1C Fig). The two haCRs targeted sites were placed in close proximity to each other (Cas9 cut sites 79 bp apart) to be able to determine frameshifts in the same sequencing analysis. haCR#1 produced frameshifts in 24 of 33 (72%) clones and haCR#2 in 20 of 33 (60%). When both sites were taken into account simultaneously, the frameshift rate was increased to 87% of the alleles (29/33 clones sequenced) (S1A–S1C Fig), which was close to the theoretical value of 89% for a single allele.

To test whether haCR targeting would produce a substantial reduction in protein function, we targeted *hexb*, which codes for the lysosomal enzyme and microglial marker hexosaminidase, with two haCRs [28], because a quantifiable enzyme assay was available to us. haCR#1 reduced enzyme activity by 63% and haCR#2 by 80%. Combining the haCRs reduced enzyme activity by 80%, not more than haCR#1 alone (S1D Fig). Hence, one haCR can already lead to strongly reduced protein function.

In summary, we have established that pre-screening of sCrRNA activity is important and can be done using a rapid RFLP-based approach. Using one or two identified haCRs can lead to substantial gene disruption. This informs the decision on striking the right balance between effectiveness of gene targeting and summation of potential off-target effects in any phenotypic screen.

## Phenotypic screening

For phenotypic analysis, we focussed on 30 potentially macrophage-related genes. Genes and background information are listed in S2 Table. Based on the above considerations, we decided to limit targeting to maximally two haCRs per gene where available and only used one haCR in cases in which we could target functionally important domains. For duplicated genes (*slc2a5*, *timp2* and *lpl*), both paralogs were targeted simultaneously with two haCRs each to avoid potential compensation by upregulation of the remaining paralog [29].

As a read-out for an impact of haCR injections on spinal cord regeneration we determined whether haCR-injected larvae showed impaired axon growth across a spinal injury site in a simple assay [3,4], presented in Fig 2A. Using animals with transgenically labelled neurons and their axons (Xla.Tubb:DsRed), we can visualise the spinal cord in side-views. Freshly injured animals and those that do not recover from the injury show a gap in fluorescence along the spinal cord (Fig 2B). At 24 hours post-injury (hpl) 40% and at 48 hpl, about 80% of larvae that have been injured at 72 hours post-fertilization (hpf) show transgenically labelled axons that grow across the injury site to re-connect the spinal cord. While we have shown recovery of swimming capacity in this paradigm by 48 hpl [3,4], full regeneration might take longer, as observed in other larval lesion paradigms [5,30].

Determining the percentage of larvae with continuity of axonal labelling across the spinal injury site (called "bridging") at 24 and 48 hpl is very rapid, a necessity in a screening

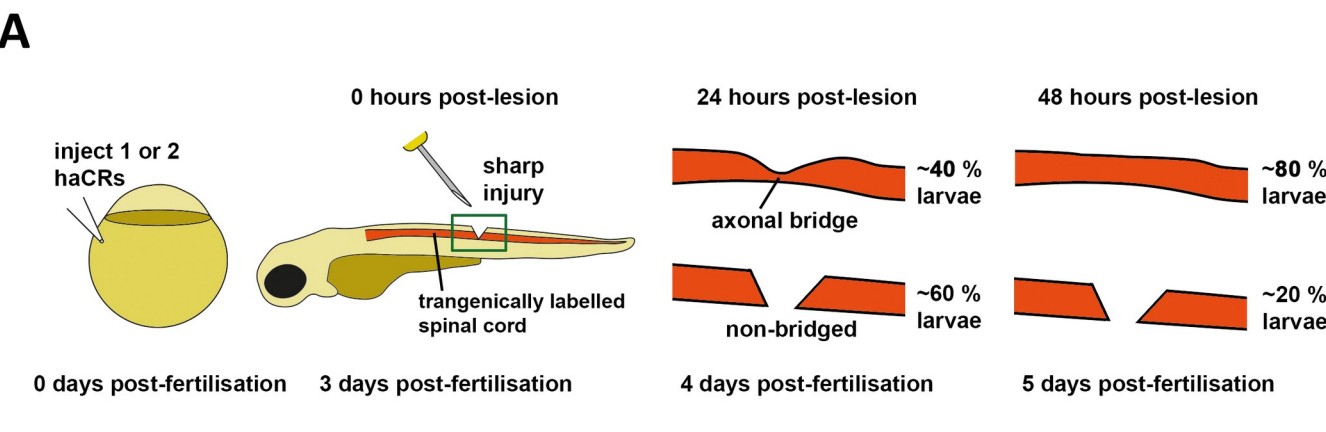

**Fig 2. Phenotypic screening reveals modifiers of spinal cord regeneration. A:** A schematic representation of the spinal cord regeneration assay. Percentages indicate expected proportions of larvae with an injury site bridged by axons in controls. **B:** Example images of unlesioned, non-bridged (star indicates gap of neuronal labeling) and bridged spinal cord (white arrow) are shown (lateral views). Scale bar = 50 μm. **C:** Results of spinal cord regeneration screen for all screened genes at 48 hpl are shown. Significant reductions in bridging, normalised to control lesioned animals, are observed for *cst7* (p < 0.0001), *sparc* (p = 0.04), *tgfb1a* (p = 0.03), *tgfb3* (p = 0.005), *tnfa* (p < 0.0001), *ifngr1* (p = 0.0013,) *hspd1* (p = 0.011), *tbrg1* (p = 0.0494, *serpinb1* (p = 0.0279), and *mertk*

(p = 0.0195); * indicates significance at 48 hpl; # indicates significance at 24 hpl (see S2 Fig); number of larvae per experiment are indicated at the bottom of each bar. For *dpm3* no viable larvae could be raised. A single sCrRNA targeting a key functional domain was used to target *ctsd*, *abca7*, *sparc*, *clip3*, *abca1b*, *tnfa*, *tgfb1a* and *tgfb3*. Two sCrRNAs were used to target all remaining genes. **D:** Mutant analysis confirms axonal phenotypes for *sparc* (p = 0.0189), *tgfb1a* (p = 0.019), *tgfb3* (p = 0.043) and *tnfa* (p = 0.024), but not for *cst7* (p = 0.079) at 48 hpl. The table compares the magnitude of effects between acute injection and in mutants. Fischer's exact test was used for all analyses.

application. At the same time this previously established scoring method is a sensitive measure to detect changes in the efficiency of axonal regeneration [3,4]. For example, we have shown that our scoring method correlates with more complex measurements, such as the thickness of the axon bridge, and with the degree of recovery of swimming function animals show after injury.

Of the 30 targeted genes, we found 10 'hits' that significantly reduced the proportions of larvae with axon bridging at 48 hpl (Fig 2C). The effect of disrupting one of these genes (*tnfa*) using acute injection of CRISPR gRNAs has previously been described [3], and thus served as a positive control.

## Validation of hit genes

To validate hit genes, we raised stable mutants for four genes of particular interest, *tgfb1a*, *tnfa*, *sparc*, and *cst7*, because we observed reduced rates of larvae with bridged injury sites at both the early (24 hpl; S2 Fig) and the late time point (48 hpl) of analysis after spinal injury for these. This indicates that these genes may be essential for regeneration from an early time point. We added a fifth hit gene (*tgfb3)* to the list for mutant validation, to better understand the relative functions of *tgfb3* and *tgfb1a* as anti-inflammatory cytokines (see Material and Methods for mutant generation from haCR-injected larvae).

In all of the stable lines, the induced mutations produced premature stop codons, confirmed by direct sequencing, and therefore were likely to abrogate gene function (S3 Fig). For phenotypic analysis, we outcrossed mutants to wildtype animals and analysed incrosses of these heterozygous animals (F3 generation) to mitigate the risk of carrying forward any potential background or off-target mutations. Proportions of larvae with bridged lesion sites were assessed against wildtype siblings.

With the exception of *cst7*, all stable mutants showed impaired axon bridging of the spinal lesion site at 48 hpl as predicted from acute studies of haCR-injected animals (Fig 2D). *tgfb1a* mutants (mutant: 79% larvae with bridged injury site compared to wildtype; acute injection: 73%; Fig 2D) and *sparc* mutants (Mutant: 71% larvae with bridged injury site compared to wildtype; acute injection: 74%; Fig 2D) showed comparable magnitudes of the bridging phenotype to acute haCR injections. *tnfa* mutants (72% mutant larvae with bridged injury site compared to wildtype; acute injection: 53%) and *tgfb3* mutants (86% mutant larvae with bridged injury site compared to wildtype; acute injection: 68%) showed a somewhat milder phenotype than acutely haCR injected larvae (Fig 2D).

To independently test the importance of Tgf-β signalling for spinal cord regeneration, we used the small molecule inhibitor SB431542 of the transforming growth factor beta receptor 1 (Tgfbr1, formerly known as Alk5), which has previously been validated in zebrafish [31]. This led to a reduction in the frequency of larvae with bridged injury sites to 75.4% of control larvae at 2 dpl (S4 Fig), similar to the reduction seen in *tgfb1a* and to a lesser extent in *tgfb3* mutants. Hence, Tgf-β signalling is necessary for unimpeded axonal regeneration.

Glial processes also bridge the spinal injury site and we wanted to know whether their growth would also be impaired by targeted genes that led to pronounced axonal phenotypes. We assessed glial bridging using immunohistochemistry for the glial fibrillary acidic protein

(Gfap) and found that only *tgfb3* mutants showed impaired bridging of glial processes, whereas *tgfb1a*, *sparc*, and *tnfa* mutants did not show significant effects on glial bridging (S5 Fig). This suggests differential roles of genes for glial and axonal regrowth.

In summary, we confirmed impaired spinal cord regeneration for four of the five hit genes tested in stable mutants and we confirmed the importance of Tgf-β signalling in two stable mutants and by pharmacological inhibition.

## *tgfb1a* controls post-injury inflammation

We decided to analyse the consequences of targeting *tgfb1a* and *tgfb3* on spinal cord regeneration in more detail, because both are expressed in macrophages in the injury site [3] and may be needed for the function of macrophages in mitigating the inflammatory response after injury. In the absence of macrophages, the initial pro-inflammatory reaction to a lesion, consisting of high numbers of neutrophils and high levels of Il-1β, persists at 48 hpl, whereas in controls both measures sharply drop a few hours after injury. Controlling Il-1β levels is crucial to allow regeneration [3].

To determine whether deficiency in *tgfb1a* or *tgfb3* would lead to a similarly prolonged inflammation, we assessed numbers of neutrophils, macrophages and levels of *il1b* expression after perturbation of *tgfb1a* and *tgfb3* at 48 hpl. To visualise macrophages in the injury site, we used the *mpeg1*:GFP transgenic line. We did not find any changes in macrophage numbers after injection of haCRs to *tgfb1a* or *tgfb3* in lesioned animals, compared to lesioned controls (Fig 3A and 3B), indicating that these genes are not necessary for macrophages to populate the injury site.

To visualise neutrophils, we used an antibody to Mpx in the same animals as above. Injection of *tgfb1a* haCRs led to a 63% higher count of neutrophils after injury compared to lesioned controls (Fig 3A and 3C). Similarly, counting neutrophils in heterozygous incrosses of *tgfb1a* mutants indicated a 66% higher number in homozygous mutants. In heterozygous larvae, a 54% higher number was observed, indicating a possible dose effect of gene copy number (Fig 3E and 3F). Higher neutrophil counts are likely a consequence of an impaired ability of macrophages to control neutrophil numbers, e.g. by inducing reverse migration of these [32]. Lesioned animals injected with *tgfb3* haCRs showed a trend towards an increase in neutrophil counts of 23% compared to lesioned controls, but this did not reach significance (p > 0.05; Fig 3A and 3C). This is consistent with the observed smaller effects of *tgfb3* mutation on axonal regrowth than those of *tgfb1b* mutation.

To measure levels of *il1b* expression, we used qRT-PCR on the isolated injured trunk region [3]. This showed a 120% higher level in *il1b* expression at 48 hpl after injection of *tgfb1a* sCrRNA, but no effect for *tgfb3* (Fig 3A and 3D), compared to control lesioned animals. The continued presence of high numbers of neutrophils and high *il1b* expression at 48 hpl after *tgfb1a* targeting indicates prolonged inflammation, shown to inhibit regeneration [3].

To directly test whether increased levels in the *tgfb1a* mutant are contributing to reduced axon bridging, we inhibited Il-1β signalling in *tgfb1a* sCrRNA-injected animals using the caspase-1 inhibitor YVAD, as described [3]. This treatment significantly improved the axon bridging phenotype (Fig 3G). Together, this indicates a role of *tgfb1a* in controlling *il1b* levels, whereas the contribution of *tgfb3* is either very low or it acts through other mechanism to support regeneration.

We also analysed touch-evoked swimming distance in *tgfb1a* mutants. However, no significant impairment was observed (S6 Fig). This may be explained by a non-linear relationship between relatively modest anatomical recovery already leading to substantial functional recovery after spinal cord injury [33].

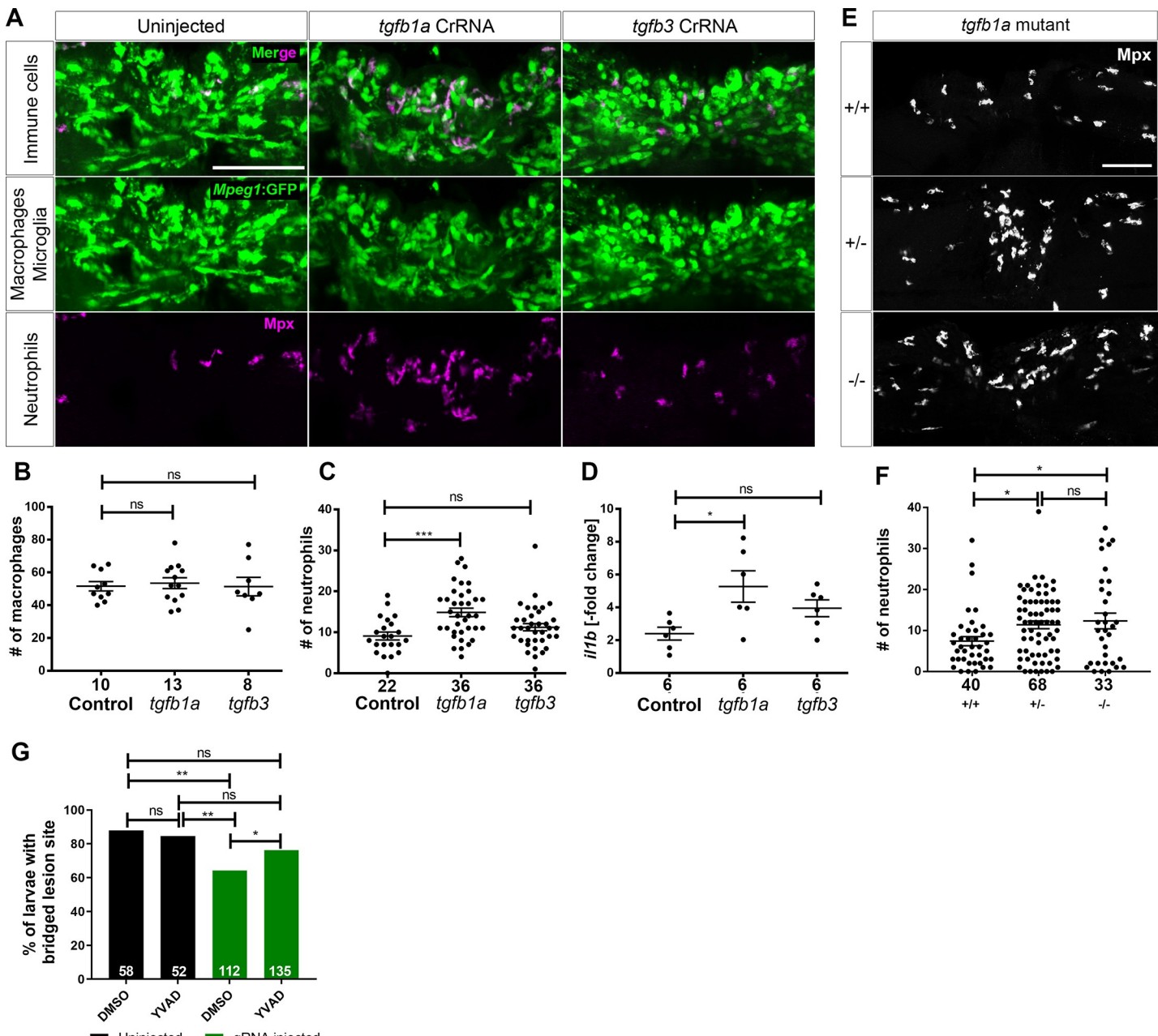

**Fig 3. Loss of *tgfb1a* leads to prolonged inflammation. A:** Lateral views of lesion sites in larval zebrafish are shown with the indicated markers and experimental conditions at 48 hpl. **B-C:** Quantifications show that numbers of macrophages were not altered by injecting any of the indicated haCRs (B; one-way ANOVA with Bonferroni's multiple comparison test; Theoretical power = 0.85 to see a similar increase as for neutrophils in C), neutrophils were increased in number in *tgf1b* haCRs injected animals (one-way ANOVA with Bonferroni's multiple comparison test, P = 0.0006), but not in *tgfb3* haCRs injected animals (p = 0.32). **D:** Animals injected with *tgfb1a* haCRs, but not those injected with *tgfb3* haCRs (P = 0.36), displayed marked increases in *il1b* expression levels in the lesion site compared to lesioned controls at 48 hpl (one-way ANOVA with Bonferroni's multiple comparison test, P = 0.0211). All transcript levels were normalized to uninjected, unlesioned controls. **E-F:** *tgfb1a* heterozygous (one-way ANOVA with Tukey's multiple comparison test, P = 0.0497) and homozygous mutant animals (P = 0.039) show increased numbers of neutrophils, comparable to haCR-injected animals. **G:** Inhibition of Il-1β with YVAD rescued axonal bridging compared to the DMSO-treated control group in animals injected with *tgfb1a* haCRs (Fisher's exact test * p <0.05, ** p<0.01). Numbers in B, C, F indicate numbers of animals; in D numbers of independent experiments. Error bars represent standard error of the mean (SEM). Scale bars = 50 μm.

In summary, our haCR screen has found several potentially macrophage-related genes to be involved in successful spinal cord regeneration in zebrafish and reveals *tgfb1a* and *tgfb3* as promoting spinal cord regeneration, with *tgfb1a*, at least in part, doing so by controlling neutrophil numbers and *il1b* expression.

## Discussion

We establish an efficient sCrRNA screening paradigm that involves a pre-screening step to compensate for inherent variability in sCrRNA activity and to identify highly active guides (haCRs) for *in vivo* phenotypic screens. We used haCRs to target 30 potentially macrophage-associated genes in a spinal cord injury assay and validated four hits through generation of stable mutant lines as playing key roles in successful spinal cord regeneration in zebrafish. We further identified *tgfb1a* as a regulator of post-injury inflammation, providing a mechanistic basis to understand how inflammation is rapidly resolved to promote recovery.

### The immune reaction is essential for spinal cord regeneration

The phenotypes of *tgfb1a* sCrRNA-injected and mutant animals with an increase in neutrophil numbers and *il1b* expression levels resembled that of a lack of macrophages [3]. In the macrophage-less *irf8* mutant, neutrophils show a slower clearance rate from the injury site after the peak at 2 hours post-lesion, such that their number is still higher relative to wildtype animals at 48 hpl. Likewise, levels of *il1b*, expressed mainly by neutrophils, peak at 4–6 hours after injury and rapidly decline thereafter, but do so more slowly in the *irf8* mutant. Crucially, phenotypes in *irf8* mutants can be rescued by inhibiting Il-1β function alone, indicating that a critical macrophage function is control of Il-1β. We could similarly rescue axon bridging in the *tgfb1a* mutant. As *tgfb1a* and *tgfb3* are typical anti-inflammatory cytokines [34] and are expressed by macrophages in zebrafish [3] and mammals [35] after spinal injury, Tgf-β signalling may thus be a part of the mechanism by which macrophages control Il-1β levels and thereby promote spinal cord regeneration. However, *tgfb1a* and *tgfb3* are also expressed by other cell types [3] and may interact with cell types other than neutrophils, such as astrocytes [36] or the neurons directly [34,37]. Since the interactions of a number of different cell types, such as immune, and neural cell types, but non-neural cells, such as fibroblasts and keratinocytes, are highly complex in the spinal injury site [2], future research will have to dissect influences of Tgf-β signalling on spatio-temporal interactions of these cell types.

Our screen found additional regeneration-relevant genes that we have confirmed in mutants. Tnfa is mainly produced by macrophages in the spinal injury site and has been shown to be necessary for larval spinal cord regeneration using pharmacological inhibition of Tnfa release and acute sCrRNA injection in zebrafish [3]. We confirm these findings here in a stable *tnfa* mutant.

Sparc is produced by macrophages and organises the collagen matrix [38]. Expression of *Sparc* by transplanted olfactory ensheathing glia in mammalian spinal cord injury has been shown to be beneficial to regeneration [39]. In zebrafish larvae, regenerating axons grow in close contact with fibrils of extracellular matrix containing growth-promoting ColXII [4]. Hence Sparc protein may contribute to this axon growth promoting matrix in zebrafish. However, *sparc* has multiple functions, for example in synapse formation [40,41], which will be analysed in future detailed phenotypic analyses of the mutant.

The above suggested mechanisms of hit genes imply indirect actions on axon regrowth by modulating the inflammatory or extracellular matrix environment. However, direct actions on expression of pro-regenerative genes in axotomized neurons is also possible. Future studies, for example using single cell RNAseq analysis, could help to distinguish between these possibilities.

Previous work has suggested that bridges of glial processes across the injury site may form a substrate for axons to cross the injury site in adult zebrafish [42,43]. However, time-lapse observation in larvae has shown that axons often grow independently of glial processes and also cross the injury site when glial processes are ablated [4]. Here we observe that most mutants do not show glial phenotypes and find that the *tgfb3* mutant exhibits a strong glial phenotype, but a relatively mild axonal phenotype. This suggests differences in glial and axonal crossing mechanisms of the injury site and that our screen is sensitive to genes that are necessary for axonal regrowth.

Our present screen has already identified some promising candidate genes that will help us to understand the pro-regenerative role of macrophages and potentially other immune cells in zebrafish. Further hits will be confirmed in stable mutants and our screening approach can be scaled up to analyse potential pro-regenerative functions in other cell types that crucially support regeneration, such as fibroblasts [4]. The superb optical accessibility of the larval zebrafish injury model will allow us to follow cell type interactions in space and time to understand the mechanistic contribution of essential genes and cell types to successful spinal cord regeneration in zebrafish.

## Pre-screening for activity improves phenotypic screening with sCrRNAs

Finding functional immune system-related genes in successful spinal cord regeneration was possible due to our novel screening paradigm, facilitated by *in vivo* pre-screening of sCrRNAs for activity. Knowing that sCrRNAs used are highly active in the specific context in which they will be used for phenotypic screening allowed us to reduce the number of sCrRNAs to maximally two pre-determined haCRs per gene, which reduces the risk of off-target effects [21] and false negative results. Although we found phenotypes in our pilot screen using only 1 haCR for certain genes, 2 haCRs might be preferable to achieve a consistent strong reduction in gene function. Furthermore, as haCRs are fairly common (44% of CrRNAs tested), identifying a pair, is not very time consuming. We cannot exclude false negative findings resulting from functional protein still being produced despite premature STOP codons being introduced. However, this is the exception rather than the rule, and generally only occurs when splice sites have been mutated [44]. A two-haCR approach will likely limit false negative findings. It is also important to state that a negative result in the highly targeted read-out for our screening assay does not mean that targeted genes are not important for other immune functions.

Pre-screening has become feasible due to the availability of synthetic CrRNAs, which have a higher likelihood of being highly active than *in vitro* transcribed sgRNAs and are more versatile in target selection [19,27,45]. However, despite the high rate of haCRs detected, activity of individual sCrRNAs is highly variable and unpredictable. It is unlikely that the enzyme recognition sites chosen for our RFLP-based sCrRNA activity are particularly resistant to CRISPR manipulations, because we used varied enzymes with unrelated sequences, yet variability in activity remained. Nevertheless, other highly efficient methods to determine sCrRNA activity, such as deep sequencing are available [46]. Lack of chemical modification of the RNA oligos is also unlikely to be responsible for variable efficacy according to findings by others [19]. Finally, experimental variations, such as injection location and vehicle composition are unlikely to decisively influence effectiveness of sCrRNAs, as described by others [22] and also reflecting our own experience. Hence, while we cannot exclude that optimisation of experimental protocols could increase haCR frequency, the above considerations suggest sCrRNA activity may be intrinsically variable. Without knowing the complex reasons for these differences, pre-screening presents a viable work-around for screening purposes.

We applied current prediction rules for CrRNA activity to our collection of sCrRNAs and found only weak correlations between predicted activity and the observed *in vivo* activity of

350 separate guides, which is to our knowledge the largest profiling of *in vivo* synthetic gRNA activity carried out to date. Although very high somatic activity of specific sCrRNAs has been demonstrated [19], only an unbiased testing of large numbers of sCrNRAs could reveal variability. Therefore, determining CrRNA activity *in vivo* by pre-screening is hugely advantageous compared to a purely *in silico* approach. We used RFLP analysis to estimate activity of sCrRNAs, because RFLP is a rapid and relatively inexpensive method. Efficiency of sCrRNAs can be exactly determined using deep sequencing, but this method produces a high amount of raw data that is impractical in a screening approach. Other methods such as T7 endonuclease assays and high-resolution melting analysis could be used, however, these may have difficulty identifying high levels of sCrRNA activity, if indel diversity is minimal or if there is a high level of identical biallelic mutations [47,48]. Our direct sequencing and protein activity assays have shown that our RFLP-based assay is capable of finding differences in sCrRNA activity and is unlikely to skew our activity measurements significantly.

Phenotypic screening of regeneration with haCRs is limited to genes with no essential developmental function. Indeed, we observed excessive mortality when targeting *dmt3*, probably because of essential developmental roles of the gene [49]. It is likely that we did not observe more instances of non-developing embryos, because we targeted mostly macrophage-related genes and macrophages are not essential for early development [50].

Our screen had a relatively high hit rate of 33%. This was expected, as genes were pre-selected for likely functions in macrophages that play a crucial role in controlling the inflammation in successful spinal cord regeneration in zebrafish [3]. We confirmed a role for the genes in spinal cord regeneration in stable mutants for 4 out of 5 genes (80%). This indicates that the screening paradigm has a relatively low rate of false positive findings. However, even with our 2-guide approach, we encountered a false positive in *cst7* and two of the other four mutants showed a weaker phenotype than after acute injection of haCRs. Despite CrRNAs having few predicted off-target effects, we cannot exclude that these exaggerated the phenotype. Sensitivity to off-target effects may also vary depending on design and read-out, which will be revealed in future screens. Overall, this highlights that guide number should be kept to a minimum and that validation of novel phenotypes with stable mutants is advised.

In conclusion, through a CRISPR based phenotypic screen in larval zebrafish, we identify genes that are crucial for successful spinal cord regeneration. A similar approach has recently been proposed for adult zebrafish [51]. We are using a specific injury paradigm as a read-out. However, our sCrRNA approach can be used in a variety of developmental or injury contexts where specific cell types may be transgenically labelled and a simple and robust read-out can be devised or be gleaned from small molecule screens in zebrafish [52]. This could be done in an automated fashion to further increase throughput [53]. Hence, the rapid phenotypic screening approach that uses sCrRNAs of pre-defined high activity *in vivo* is versatile and can be adapted to any biological context of interest.

## Material and methods

### Ethics statement

All experiments were reviewed by the Animal Welfare and Ethical Review Body at the University of Edinburgh and approved by the British Home Office (project license no.: 70/8805).

### Animal husbandry

All zebrafish lines were kept and raised under standard conditions. The following lines were used: WIK wild type zebrafish [54], Tg(Xla.Tubb:DsRed)[zf14826], abbreviated as Xla.Tubb:DsRed [55]; Tg(*mpeg1*:EGFP)[gl22], abbreviated as *mpeg1*:GFP [56].

## Crispr/Cas9 design and injection

All sCrRNAs were designed so a restriction enzyme recognition sequence overlapped the Cas9 cut site. sCrRNAs and TracrRNA were purchased from Merck KGaA (Germany, Darmstadt). Cas9 (M0386M, NEB, Ipswich USA) and all restriction enzymes were purchased from NEB (Ipswich USA). Cas9 was diluted to 7 µM with diluent buffer B (NEB, Ipswich USA) on arrival and stored at -20˚C. RNA oligos were re-suspended to 20 µM with nuclease free water, and stored at -20 until use. For *in vivo* testing of CrRNA activity, 1 nl of an injection mixture composed of 1 µl of each sCrRNA (up to 4), 1 µl TracrRNA, 1 µl Cas9 and 1 µl Fast Green (Alfa Aesar, Heysham, UK), was injected into the yolk of single cell stage embryos.

## Generation of stable mutants

CrRNAs targeting exon 1 were injected into the yolk of one-cell stage embryos. The CrRNA target sites for *tgfb1a* and *tgfb3* were 5' ATGGCTAAAGAGCCTGAATCCGG and 5'GAATC CATCCAGCAGATCCCTGG, respectively. *tnfa* was targeted with 5' ACAAAATAAATGC CATCATCGGG, *sparc* with 5' CTAAACCATCACTGCAAGAAGGG and *cst7* with two sCrRNAs, 5'TTCTGCAGAGCTCCTGGGATCGG and 5' TAAAAGAGTAAGTTCCAGT CAGG, to generate a larger deletion. Founders were identified and out crossed to WT (F1), then crossed to WT again to generate the F2 generation. F2 heterozygous individuals were crossed to Xla.Tubb:DsRed or wildtype to generate the F3 generation. All spinal cord lesion assays were performed on an F3 heterozygous incross.

The *tgfb1a* line was genotyped with primers F 5' GATTTGGAGGTGGTGAGGAA and R 5' TCGCTCAGTTCAACAGTGCTAT. The *tgfb3* line was genotyped with primers F 5' GGG TCAGATCCTCAGCAAAC and R 5' GAGATCCCTGGATCATGTTGA, the *sparc* line with F 5' TGCCTAAACCATCACTGCAA and R 5' ATGCTCGAAGTCTCCGATTG, and the *cst7* line with F 5' TTGTGTGCTCTTTGCTGTCTG and R' CTGCACCTGTCTCTTTGCAC. The *tnfa* line was genotyped with primers F 5'ACCAGGCCTTTTCTTCAGGT and R 5' AGCGG ATTGCACTGAAAAGT followed by a *bstXI* digest.

## RFLP analysis

At 24 hpf, DNA from single embryos was extracted using 100 µl of 50 mM NaOH and 10 µl of Tris-HCl pH 8.0 as previously described [57] All RFLP analyses were conducted on DNA of separate individuals and not pooled in order to accurately determine mutation rate. For each CrRNA, RFLP was conducted on 4 uninjected controls and 8 injected individuals. Complete digestion in controls served as indicator of appropriate gel detection sensitivity, enzyme activity, and purity of DNA.

PCR products were generated with BIOMIX red (BIOLINE, London, UK) and 1 µl of the respective restriction enzyme was added directly to the final PCR product for RFLP analysis, without the addition of extra buffers and incubated at the optimal temperature for each respective enzyme. 20 µl of digest were run on 2% agarose gel (BIOLINE, London, UK) and imaged on a trans illuminator. Band intensities were calculated using imageJ (http://imagej.nih.gov/ij).

## Allele sequencing

PCR products from 8 injected embryos (24 hpf) were pooled and ligated into a StrataClone vector and transformed into competent cells, following the manufacturer's instructions (Agilent, Santa Clara, USA). Positive colonies were identified and sequenced using M13 primers.

## Hexb activity assay

Hexb activity was determined as previously described [58]. Briefly, 3 dpf embryos, 20 per clutch, 3 independent clutches, were homogenised in 100 µl of nuclease free water. Each sample was diluted 1/10 with McIlvaine citrate–phosphate buffer pH 4.5 and activity assayed with 4-methylumbelliferyl-2-acetamido- 2-deoxy-β-D-gluco-pyranoside (Sigma, Dorset, UK) in a plate reader.

## Immunohistochemistry on whole-mount larvae

All incubations were performed at room temperature unless stated otherwise. At the time point of interest, larvae were fixed in 4% PFA-PBS containing 1% DMSO at 4˚C overnight. After washes in PBS, larvae were washed in PBTx. After permeabilization by incubation in PBS containing 2 mg/ml Collagenase (Sigma) for 25 min larvae were washed in PBTx. They were then incubated in blocking buffer for 2 h and incubated with primary antibody (anti-MPX, GeneTex GTX128379, Irvine, California, USA) diluted in blocking buffer at 4˚C overnight. On the following day, larvae were washed 3 x in PBTx, followed by incubation with secondary antibody diluted in blocking buffer at 4˚C overnight. The next day, larvae were washed three times in PBTx and once in PBS for 15 min each, before mounting in 70% glycerol.

For whole mount immunostaining of acetylated tubulin (Sigma T6793) and GFAP (Dako Z0334) to visualise the axons and the glial processes, respectively, larvae were fixed in 4% PFA for 1 h and then were dehydrated in 25% 50%, 75% MeOH in 0.1% Tween in PBS, transferred to 100% MeOH and then stored at -20˚C overnight. The next day, head and tail were removed, and the samples were incubated in pre-chilled 100% acetone at -20˚C for 10 min. Thereafter, larvae were washed and digested with Proteinase K (10 µg/ml) for 15 min at room temperature and re-fixed in 4% PFA. After washes, the larvae were incubated with 4% BSA in PBTx for 1 h. Subsequently, the larvae were incubated over two nights with primary antibodies (anti-acetylated tubulin, anti-GFAP). After washes and incubation with the secondary antibody, the samples were washed in PBS for 15 min each, before mounting in glycerol.

## Spinal cord injury

At 3 dpf, zebrafish larvae were anaesthetised in PBS containing 0.02% aminobenzoic-acid-ethyl methyl-ester (MS222, Sigma), as described [3]. Larvae were transferred to an agarose-coated petri dish. Following removal of excess water, the larvae were placed in a lateral position, and the tip of a sharp 30.5 G syringe needle (BD Microlance) was used to inflict a stab injury or a dorsal incision on the dorsal part of the trunk at the level of the 15th myotome, leaving the notochord intact.

## Compound incubation

SB431542 (Abcam, ab120163) was dissolved in DMSO to a stock concentration of 100 mM and added to the larvae as indicated at a final concentration of 50 µM. The final concentration of DMSO was 1% and the same concentration of DMSO was added to the controls. Larvae were pre-treated for 2 hours before the injury and were incubated with the drug for 48 hours.

Ac-YVAD-cmk (YVAD) (Sigma, SML0429) was dissolved in DMSO to a stock concentration of 10 mM. The working concentration was 50 µM prepared by dilution from the stock solution in fish water. Larvae were pre-treated for 2 h before the injury and were incubated with the drug for 48 hpl as previously described [3]. Axonal and glial bridging was assessed blinded to the experimental condition on three independent clutches of larvae.

## Behavioural analysis

Behavioural analysis was performed as previously described [59]. Lesioned larvae were touched caudal to the lesion site using a glass capillary. The swim distance of their escape response was recorded for 15 s after touch and analyzed using a Noldus behaviour analysis setup (EthoVision version 7). Data given is averaged from triplicate measures per fish. Between repeated measures, the larvae were left to recover for 30 sec. The observer was blinded to the treatment during the behavioural assay.

## Assessment of axonal and glial phenotypes

Re-established axonal connections and glial connections ("bridges") were scored at the time point of interest in fixed immunolabelled samples (for axonal and glial bridges) and live transgenic animals (for axonal bridges). Larvae were directly visually evaluated using a fluorescent stereomicroscope (Leica M165 FC) or confocal imaging (Zeiss LSM 710, 880). Larvae were scored as described [3] with the observer blinded to the experimental condition. Briefly, a larva was scored as having a bridged lesion site when continuity of the axonal labeling between the rostral and caudal part of the spinal cord was observed. The same criterion was used for the assessment of the glial bridges. Continuity of labeling was defined as at least one fascicle being continuous between rostral and caudal spinal cord ends, irrespective of the fascicle thickness. Larvae in which the lesion site was obscured by melanocytes or the notochord was inadvertently injured were excluded from the analysis.

## Quantitative RT-PCR

Reverse transcription of 500 ng RNA was performed with the iSCRIPT kit (BIORAD, Hercules, USA). Standard RT-PCR was performed using 10 mM of each dNTP and each primer. qRT-PCR was performed at 58°C using Roche Light Cycler 96 and relative mRNA levels determined using the Roche Light Cycler 96 SW1 software. Samples were run in duplicates and expression levels were normalized to a β-actin control. Primers were designed to span an exon–exon junction using the Primer-BLAST (https://www.ncbi.nlm.nih.gov/tools/primer-blast/) software.

## Assessment of immune cell numbers

A volume of interest was defined centered on the lesion site from confocal images. The dimensions were: width = 200 μm, height = 75 μm (above the notochord), depth = 50 μm. Images were analysed using the Imaris (Bitplane, Belfast, UK) or the ImageJ software. The number of cells was quantified manually in 3D view, on at least three independent clutches of larvae, blinded to the experimental condition.

## Experimental design and statistical analysis

No formal randomization method was used. All mutant analyses were performed on incrosses of heterozygous animals without prior knowledge of genotype. For all experiments and analyses the experimenter was blinded to the experimental condition. Image analysis was performed using ImageJ. Power analysis using G*Power (Faul et al. 2009), was used to calculate power (aim > 0.8) for the experiments and determine the group sizes accordingly. Statistical power was > 0.8 for all experiments. All quantitative data were tested for normality and analyzed with parametric and non-parametric tests as appropriate. The statistical analysis was performed using IBM SPSS Statistics 23.0. Shapiro-Wilk's W-test was used in order to assess the normality of the data. Kruskal-Wallis test followed by Dunn's multiple comparisons, One-way

ANOVA followed by Bonferroni multiple comparisons test, two-way ANOVA, followed by Bonferroni multiple comparisons, t-test, Mann–Whitney U test or Fischer's exact test were used, as indicated in the figure legends. $^{*}P< 0.05$, $^{**}P< 0.01$, $^{***}P< 0.001$, n.s. indicates no significance. Error bars indicate the standard error of the mean (SEM). The figures were prepared with Adobe Photoshop CC and Adobe Illustrator CC. Graphs were generated using GraphPad Prism 7.

## Supporting information

**S1 Fig. Injecting two haCRs simultaneously effectively disrupts gene function. A-C:** Direct sequencing of mutant alleles in embryos injected with two haCRs per gene demonstrates induction of frameshift frequencies of 72% (A; haCr #1) and 60% (B; haCR #2), when either sCrRNA is analysed individually. This rises to 87% when frame-shift frequencies are combined (C). **D:** At the protein level, dual haCR injection against *hexb* reduces enzyme activity by 80% *in vivo* (ANOVA with Tukey post-test; $p < 0.0001$). Single haCRs also reduce enzyme activity ($p < 0.0001$ for both). haCR#2 is more efficient than haCR#1 ($p = 0.0051$). Error bars represent SEM.
(TIF)

**S2 Fig. Inhibition of axon bridging is observed for some of the selected genes at 24 hpl.** Significant reductions in bridging were detected after acute injection of haCRs for *sparc* ($p = 0.0424$), *tnfa* ($p = 0.0068$), *tgfb1a* ($p = 0.0225$) and *cst7* ($p = 0.0418$). Fisher's Exact test, $p<0.05$.
(TIF)

**S3 Fig. Mutations likely lead to non-functional protein.** Deletions are shown in red and insertions are shown in blue. All stable mutations produce frameshifts (*cst7*, *tnfa*, *tgfb1a* and *sparc*) and premature stop codons, with the exception of the mutation in the *tgfb3* gene. The latter contains an in-frame indel in which the large quantity of inserted material contains a nonsense mutation.
(TIF)

**S4 Fig. Inhibiting Tgf signalling with SB31442 impairs axonal bridging.** Experimental timeline, lateral views of embryos and quantification are shown. Fisher's Exact test, $p<0.05$. The white arrow indicates the axonal bridge. Scale bar 50 μm.
(TIF)

**S5 Fig. Glial bridging is impaired in the *tgfb3* mutant.** Lateral views of whole-mounted larvae at 5 dpf are shown; rostral is left, dorsal is up. **A:** Gfap immunohistochemistry shows longitudinal processes over the injury site (centred in right column) for wildtype (WT), *sparc*, *tnfa*, and *tgfb1a* mutants, but not for *tgfb3* mutants at 48 hours post-injury (hpf). **B:** Quantification of the phenotypes shows a significant reduction in the proportion of larvae with glial bridging only for *tgfb3* mutants (Fisher's Exact test).
(TIF)

**S6 Fig. Functional recovery is not significantly impaired in the *tgfb1a* mutant.** No change is observed in the distance that *tgfb1a* mutants swim after a touch evoked stimulus (Unpaired t test, $p = 0.2590$).
(TIF)

**S1 Table. List of sCRNAs.** This is a list all sCrRNAs used in this study and their efficiencies as determined by RFLP.
(XLSX)

**S2 Table. Genes used in spinal cord injury screen.** This is a list of genes that are used in the spinal cord injury *in vivo* screen, also containing information on expression and regulation after injury.
(XLSX)

## Acknowledgments

We thank Viviane Schulz for help with some experiments.

## Author Contributions

**Conceptualization:** Marcus Keatinge, Themistoklis M. Tsarouchas, Catherina G. Becker, David A. Lyons, Thomas Becker.

**Data curation:** Marcus Keatinge, Themistoklis M. Tsarouchas.

**Formal analysis:** Marcus Keatinge, Themistoklis M. Tsarouchas.

**Funding acquisition:** Catherina G. Becker, David A. Lyons, Thomas Becker.

**Investigation:** Marcus Keatinge, Themistoklis M. Tsarouchas, Tahimina Munir, Nicola J. Porter, Juan Larraz, Catherina G. Becker, David A. Lyons, Thomas Becker.

**Methodology:** Marcus Keatinge, Themistoklis M. Tsarouchas.

**Project administration:** Catherina G. Becker, David A. Lyons, Thomas Becker.

**Resources:** Davide Gianni, Hui-Hsin Tsai.

**Supervision:** Catherina G. Becker, David A. Lyons, Thomas Becker.

**Validation:** Marcus Keatinge, Themistoklis M. Tsarouchas.

**Visualization:** Marcus Keatinge, Themistoklis M. Tsarouchas, Catherina G. Becker, David A. Lyons, Thomas Becker.

**Writing – original draft:** Marcus Keatinge, Themistoklis M. Tsarouchas, Catherina G. Becker, David A. Lyons, Thomas Becker.

**Writing – review & editing:** Marcus Keatinge, Themistoklis M. Tsarouchas, Tahimina Munir, Nicola J. Porter, Juan Larraz, Davide Gianni, Hui-Hsin Tsai, Catherina G. Becker, David A. Lyons, Thomas Becker.

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
