## [Decision Letter · Decision Letter 0]

5 Dec 2020

Dear Dr Becker,

Thank you very much for submitting your Research Article entitled 'CRISPR gRNA phenotypic screening in zebrafish reveals pro-regenerative genes in spinal cord injury' to PLOS Genetics. Your manuscript was fully evaluated at the editorial level and by independent peer reviewers. The reviewers appreciated the attention to an important problem, but raised some substantial concerns about the current manuscript. Based on the reviews, we will not be able to accept this version of the manuscript, but we would be willing to review again a much-revised version. We cannot, of course, promise publication at that time.

If you decide to revise the manuscript for further consideration at PLOS Genetics, please aim to resubmit within the next 60 days, unless it will take extra time to address the concerns of the reviewers, in which case we would appreciate an expected resubmission date by email to plosgenetics@plos.org.

[LINK]

We are sorry that we cannot be more positive about your manuscript at this stage. Please do not hesitate to contact us if you have any concerns or questions.

Yours sincerely,

Cecilia Moens

Associate Editor

PLOS Genetics

Gregory Barsh

Editor-in-Chief

PLOS Genetics

Reviewer's Responses to Questions

**Comments to the Authors:**

Reviewer #1: In this manuscript the authors describe a method for more selecting more highly active GRISPR guides and functionally test them in the context of larval zebrafish regeneration. The method for selecting and testing the guides seems robust and applicable to organisms with a sequenced and very well annotated genome. The manuscript is well written but in some cases assumes the reader is very familiar with previous research in the field, the manuscript should be revised to ensure a reader can understand the manuscript without having to go read other papers. Specific points:

1. The authors select a list of genes to test with their new guide design method which they say are “macrophage related genes based on previous research”. Table S1 should be revised to include specific information about these genes, reference papers, have they been previously knocked out, what is the phenotype, what method was used, morpholinos, mutants, talens etc.

2. Information should be given in the text or Table S1 as to where these genes are normally expressed in development and is there an embryonic phenotype from knocking them out in the embryo?

3. Are the genes they selected normally up-regulated after spinal cord injury, is this why they were selected?

4. The authors use an axon regeneration assay using the Tubb:DsREd transgenic line to assay axon regeneration. Previous work from others in the field have shown that the glial cells must first migrate and form a glial bridge and then axon regeneration begins. It would be interesting to know if there are also defects in the formation of the glial bridge in mutants where axon regeneration is impaired.

5. Five genes which gave phenotypes were raised as stable mutants. Again it would be helpful to know where/which cell types these genes are normally expressed in after spinal cord injury. This might shed more light on why KO of tgfb1a and tgfb3 do not result in any change in the number of macrophages at the injury site, when increased number of neutrophils are observed, do they stay at the injury site for longer than in WT animals, the same with animals where an increase in macrophages in seen?

6. Overall the assay of axon regeneration and effect on immune cells are quantified at very early timepoints, even in young larvae axon regeneration takes more than 48hrs, previous work from Briona and Dorsky report axon regeneration to take up to 9 days in 4 day old larvae, while work from the Saude group reports up to 6dpl. These groups and others work on spinal cord regeneration in zebrafish, it would be good to see references to other people work in the field in this manuscript.

Reviewer #2: This study screened for macrophage genes implicated in spinal cord injury in zebrafish larvae. Recently reported high efficiency CRISPR methods were used. 30 macrophage genes were targeted. Phenotypes were scored for the presence or absence of an axon reporter bridge at the lesion site. Phenotypes were observed and confirmed in stable lines for tgfb1a tgfb3 tnfa and sparc. The study proposes that tgfb1a deletion leads to a prolonged immune response that inhibits regeneration after injury.

1- One concern is that the described screening methods report low targeting efficiencies relative to previously published studies in the field. In this study, 44% of all tested RNAs were high efficiency. This is in contrast to findings from Hoshijima et al, which convincingly reported bi-allelic indel mutations in almost all cells of the developing embryos, a result that has been recapitulated by numerous other labs. The authors do not discuss or address the reasons behind limited targeting efficiency, and instead propose pre-screening for high efficiency guide RNA to overcome this limitation.

a. The design strategy was limited to targeting enzyme recognition sites that were later used for restriction fragment length polymorphism (RFLP) genotyping. Is this limitation likely to limit targeting efficiency? Especially considering that enzyme-independent assays are now available to accurately quantify the rate of mutagenesis.

b. Compensatory mechanisms associated with targeting early exons have been described in great detail in the field but are not addressed in this study. Mutations with premature stop codons are not always likely to abrogate gene function as stated in the manuscript. Added to this concern is that all 4 phenotypes reported in this study were attributed to genes in which functional domains rather than 5’ exons were targeting.

c. A number of genes that are essential for macrophage function do not show phenotypes. Is this due to targeting strategies? Do stable p2yr1 mutants shown normal recovery for example?

d. Is it possible that the source of synthetic RNA is a contributing factor. Unclear whether the RNAs used here are chemically modified to enhance their resistance to degradation the way Alt-R-modified RNAs were shown to be.

2- The more exciting and novel aspect of this study is to apply recent CRISPR advances in zebrafish genetics to identify new genes involved in spinal cord injury. Unfortunately, the inflammatory factors tgfb1 and 3 are well characterized in spinal cord injury, and the data presented seems preliminary. For example, this reviewer finds sparc phenotypes intriguing and novel. Including expression and phenotypic data for sparc can add the missing novelty to the study. Keeping in mind that Sparc is also a known regulator of synapse function, it is important to distinguish between possible functions in neurons or macrophages.

3. The rationale behind which macrophage-related genes were included is not described in the text. it will be helpful to describe their expression and previous implication in macrophage activation.

4. Phenotyping was based on scoring an axon reporter line for bridging at the lesion site. This method does not distinguish phenotypes related to degeneration and debris clearing from regeneration phenotypes. The authors have previously reported numerous methods to assess anatomical and behavioral regeneration. We recommend using these assays for in depth characterization of the phenotypes.

As an example, Figure 3 gives a brief overview of the phenotype in tgfb mutants. Quantifications in 3b and 3c suggest that macrophage number were not changed but that tgfb1a mutants have more neutrophils. It is possible this difference is statistical since many more replicates are quantified in 3c.

Reviewer #3: The manuscript by Keatinge et al. describes a very valuable and useful scalable in vivo phenotypic screen based on CRISPR technology using imaging-friendly zebrafish larvae.

The screen was designed for a specific spinal cord injury paradigm, but I would like to emphasize the possibility of using the same screening logic to identify regulators of other biological processes, as long as they are easy to phenotype in medium-large scale.

As an important note for the general scientific community, the authors claim that the 4 different in silico prediction tools they used did not have enough prediction power to selected highly efficient guide RNASs. They propose the use of an easy and efficient RLFP-based tool and further show its efficacy in the prediction of guide RNAs.

I have a few points that I would like the authors to address.

# 1. The screening platform was designed to screen for macrophage-related genes and was successful in identifying potential macrophage players, with potential inflammation modulation effects and with potential positive effects on spinal cord regeneration. I would like to emphasize the word potential because at this stage of the work that's what the identified genes are. I think the authors should lower the tone throughout the manuscript about the involvement of these genes specifically with macrophages (as no co-stainings were done) and with regeneration capacity (as no swimming recovery test of any type was performed, only the presence of tubulin:GFP at the injury site was roughly quantified).

# 2. In Fig 1A, why do the Uninjected Digested lanes of irak3 gene have only one band? If the embryos were not injected, they should have the restriction site intact and when digested should then produce at least 2 bands. Please clarify.

# 3. In Fig 2A, I think the representation of axonal bridge at 24 and 48 hpl is misleading, as it is expected that at 48 hpl the axonal bridge at the injury site would be bigger when compared to the one seen at 24 hpl.

# 4. In Fig 2B, the examples of transgenic larvae shown are at which time point after lesion?

I find bridging a weak and limited item to access in this context. You can find larvae with a lot or very little bridging and these will be included in the same bridged category. In their quantification methodology, a larvae with only on fascicle being continuous between rostral and caudal spinal cord ends will be included in the bridged category. And the question I would like to raise at this point is to what extent can we came assumptions on the regenerative capacity based on only one fascicle crossing the injury site? A swimming test would be essential to produce a stronger conclusion.

# 5. In Fig 2C, the y axis should not be % of larvae with bridged lesion because theses values were normalized to the WT control (in which according to values in Fig 2A a represented approx. 80% and not 100% as seen in the 2C graphic.

# 6. In Fig 2D, why is there a discrepancy between the bridging % tendency between acute injections and stable mutations? The difference expected would be that in stale mutants the effect would be more pronounced than in the acute mosaic injections, why is this? and did they authors also see the defects at 24 hpl that they saw in the acute injected ones?

# 7. In Fig 2D, the legend says that the sparc mutant was analysed at 24 hpl but the text gives the idea that the result is from 48 hpl. Is the result the same for 24 and 48 hpl? Could the stronger phenotype of the sparc mutants be a result of the earlier timepoint analysed?

# 8. The authors when on to characterize in more detail the tgfb1a mutant and propose that Tgfb1a signaling may be part of the mechanism by which macrophages control iL1b levels and thus promote regeneration. To support this hypothesis the authors could simply do a IL1b inhibition rescue experiment similarly to what they already did for the tnfa mutant. And I would argue that it would be fundamental to complement these experiments with swimming recovery tests (tgfb1a mutant and tgfb1a mutant after rescue).

# 9. Regarding the genotyping of the stable mutants used for the phenotypic analysis (Fig 2D), it is not clear in the methods how this was done. Do the authors use individual larvae to extract DNA after doing the injury and then genotype by PCR?

**Have all data underlying the figures and results presented in the manuscript been provided?**

Reviewer #1: Yes

Reviewer #2: Yes

Reviewer #3: Yes

PLOS authors have the option to publish the peer review history of their article (what does this mean?). If published, this will include your full peer review and any attached files.

Reviewer #1: No

Reviewer #2: No

Reviewer #3: No

---

## [Decision Letter · Decision Letter 1]

28 Mar 2021

Dear Dr Becker,

We are pleased to inform you that your manuscript entitled "CRISPR gRNA phenotypic screening in zebrafish reveals pro-regenerative genes in spinal cord injury" has been editorially accepted for publication in PLOS Genetics. Congratulations! 

Before your submission can be formally accepted and sent to production you will need to complete our formatting changes, which you will receive in a follow up email. You may also consider addressing the comment of reviewer 4 about clarifying the mechanism by which macrophages regulate regeneration at that time. Please be aware that it may take several days for you to receive this email; during this time no action is required by you. Please note: the accept date on your published article will reflect the date of this provisional acceptance, but your manuscript will not be scheduled for publication until the required changes have been made.

Yours sincerely,

Cecilia Moens

Associate Editor

PLOS Genetics

Gregory Barsh

Editor-in-Chief

PLOS Genetics

Comments from the reviewers (if applicable):

Reviewer's Responses to Questions

**Comments to the Authors:**

Reviewer #1: The authors have addressed all of the reviewers comments and in doing so improved the manuscript and made it acceptable for publication.

Reviewer #3: I have no further comments.

Reviewer #4: In this MS, the Becker group develop a novel phenotypic screening paradigm for identifying macrophage-related pro-regenerative genes after spinal cord injury, using an elegant larval zebrafish regeneration model and CRISPR-based technologies. In an innovative approach, they used synthetic RNA Oligo CRISPR guide RNAs (sCrRNAs) and pre-screening for high activity in vivo; testing of 350 sCrRNAs; targeting 30 genes of interest; validating 5 candidates in loss of function studies and identifying 4 genes as positive regulators of successful regeneration in larval zebrafish. This novel screening methodology represents a valuable new tool for the field of regenerative neurobiology. The potential for wider application of this technology in other contexts is also important, such as other injury models and/or other cell types (and cell type interactions) related with injury and repair; and also, neurodevelopmental systems and neurogenesis and pathfinding studies where zebrafish are commonly used.

Having not been one of the original reviewers, I took into consideration the previous review comments and responses and read these carefully alongside the manuscript. I have no technical comments to add on top of the previous reviewers, it seems the authors have done an excellent job in addressing all of the main concerns. This study is elegantly conducted, and the findings are robust. I ask only for clarification (which could be added to the discussion) on one general point relating to their conclusions on understanding the mechanistic basis of how macrophages regulate regeneration:

In relation to the loss of function/mutation studies, can the authors comment on whether they are preventing neuronal regeneration directly (i.e., directly affecting pro-regenerative gene expression in injured neurons) or are they preventing regeneration indirectly (i.e., because there is no resolution of inflammation, the environment remains pro-inflammatory/toxic and inhibitory to growth). This is an important point in terms of mechanistic understanding. Future studies (e.g., scRNA-seq) involving assessments of gene expression in regenerating vs non-regenerating neurons could be important for studying directly pro-regenerative vs anti-inflammatory/pro-resolving mechanisms.

**Have all data underlying the figures and results presented in the manuscript been provided?**

Reviewer #1: Yes

Reviewer #3: Yes

Reviewer #4: Yes

PLOS authors have the option to publish the peer review history of their article (what does this mean?). If published, this will include your full peer review and any attached files.

Reviewer #1: No

Reviewer #3: No

Reviewer #4: **Yes: **Elizabeth Bradbury

**Data Deposition**

http://datadryad.org/submit?journalID=pgenetics&manu=PGENETICS-D-20-01501R1

**Press Queries**

---

## [Editor Report · Acceptance letter]

8 Apr 2021

PGENETICS-D-20-01501R1 

CRISPR gRNA phenotypic screening in zebrafish reveals pro-regenerative genes in spinal cord injury 

Dear Dr Becker, 

We are pleased to inform you that your manuscript entitled "CRISPR gRNA phenotypic screening in zebrafish reveals pro-regenerative genes in spinal cord injury" has been formally accepted for publication in PLOS Genetics! Your manuscript is now with our production department and you will be notified of the publication date in due course.

With kind regards,

Katalin Szabo

PLOS Genetics

On behalf of:
